# Genomic Dissection and Diurnal Expression Analysis Reveal the Essential Roles of the PRR Gene Family in Geographical Adaptation of Soybean

**DOI:** 10.3390/ijms23179970

**Published:** 2022-09-01

**Authors:** Peiguo Wang, Liwei Wang, Lixin Zhang, Tingting Wu, Baiquan Sun, Junquan Zhang, Enoch Sapey, Shan Yuan, Bingjun Jiang, Fulu Chen, Cunxiang Wu, Wensheng Hou, Shi Sun, Jiangping Bai, Tianfu Han

**Affiliations:** 1Department of Crop Genetics and Breeding, College of Agronomy, Gansu Agricultural University, Lanzhou 730070, China; 2MARA Key Laboratory of Soybean Biology (Beijing), Institute of Crop Sciences, The Chinese Academy of Agricultural Sciences, 12 Zhongguancun South Street, Beijing 100081, China; 3Council for Scientific and Industrial Research (CSIR)-Oil Palm Research Institute, Kade P.O. Box 74, Ghana

**Keywords:** soybean, photoperiodic flowering, *PRR* gene family, haplotype, geographical adaptation

## Abstract

Pseudo-response regulator (PRR) family members serve as key components of the core clock of the circadian clock, and play important roles in photoperiodic flowering, stress tolerance, growth, and the development of plants. In this study, 14 soybean PRR genes were identified, and classified into three groups according to phylogenetic analysis and structural characteristics. Real-time quantitative PCR analysis revealed that 13 *GmPRRs* exhibited obvious rhythmic expression under long-day (LD) and short-day (SD) conditions, and the expression of 12 *GmPRRs* was higher under LD in leaves. To evaluate the effects of natural variations in *GmPRR* alleles on soybean adaptation, we examined the sequences of *GmPRRs* among 207 varieties collected across China and the US, investigated the flowering phenotypes in six environments, and analyzed the geographical distributions of the major haplotypes. The results showed that a majority of non-synonymous mutations in the coding region were associated with flowering time, and we found that the nonsense mutations resulting in deletion of the CCT domain were related to early flowering. Haplotype analysis demonstrated that the haplotypes associated with early flowering were mostly distributed in Northeast China, while the haplotypes associated with late flowering were mostly cultivated in the lower latitudes of China. Our study of PRR family genes in soybean provides not only an important guide for characterizing the circadian clock-controlled flowering pathway but also a theoretical basis and opportunities to breed varieties with adaptation to specific regions and farming systems.

## 1. Introduction

Soybean (*Glycine max*) is a typical short-day (SD) plant showing photoperiod sensitivity [1,2,3,4,5]. Photoperiodic regulation of flowering time is an important determinant for the adaptability and productivity of soybean. Various efforts have been made to study the genetic and molecular mechanisms underlying photoperiodic flowering time regulation in soybean [6,7]. The molecular mechanisms of the major genes *E1* [8], *E2* [9], *E3* [10], *E4* [11], *E6* [12], *GmFT1a* [13], *GmFT2a* [14,15,16,17], *GmFT4* [18], *GmFT5a/qDTF*-*J1* [16,19], *J* [20,21,22], *qFT12*-*1*/*GmPRR37*/*Tof12*/*GmPRR3b* [23,24,25,26], *GmPRR3a*/*Tof11* [25,26] and *GmLHYs* [27,28,29] have been identified and characterized. Recent studies have demonstrated that circadian clock genes act upstream of the legume-specific flowering repressor *E1* to regulate soybean photoperiodic flowering [24,25,26,27,28,29]. Therefore, dissecting the genomic basis of the natural variation of circadian clock genes would facilitate elucidating the genetic networks of photoperiodic flowering and adaptation in soybean.

As key components of the circadian clock, pseudo-response regulator (PRR) family genes play important roles in flowering, stress tolerance, and growth in plants [30,31,32]. In *Arabidopsis*, PRR family genes consist of five members (*APRR1*/*TOC1*, *APRR3*, *APRR5*, *APRR7* and *APRR9*), and are subject to a circadian rhythm at the transcriptional level [33,34]. Comprised of the amino acid sequence of Arabidopsis *PRRs*, which are defined as proteins containing an N-terminal pseudo receiver (PR) domain and a C-terminal CCT (*CONSTANS*, *CO-like*, and *TOC1*) domain, PRR homologues in plants are classified into three groups (*PRR1*, *PRR7*/*3* and *PRR5*/*9*) [31,33].

Arabidopsis *PRRs* have been shown to regulate flowering via the photoperiod pathway [35]. In contrast, numerous studies have identified *PRR3*/*7* homologues that regulate photoperiod flowering in rice [36,37,38], wheat [39], barley [40], sorghum [41] and soybean [24,25,26]. In soybean, natural variants in *GmPRR37*/*GmPRR3b* and *GmPRR3a* affect photoperiodic flowering and have been selected during soybean domestication and facilitated the adaptation of soybean to different cultivated regions [24,25]. To date, there are few studies on other PRR family members in soybean.

In the present study, we studied 14 PRR family members of soybean and analyzed their gene structure, evolutionary characteristics, expression pattern under different photoperiodic conditions, and flowering phenotypes across six environments and geographical distributions within the major haplotypes. This study lays a foundation for the further functional characterization of specific genes in the *PRR* gene family of soybean, which will facilitate the breeding of varieties with better adaptability. 

## 2. Results

### 2.1. Genome-Wide Identification of the PRR Gene Family in Soybean

In order to identify the *PRR* gene family in soybean, the known Arabidopsis PRR protein sequences were used as queries in BLASTp searches followed by HMMER profiles. Finally, a total of 14 soybean PRR family genes were confirmed by domain analysis using Pfam and CDD tools. The 14 *GmPRR* genes were distributed on 11 chromosomes and one scaffold (Appendix A). One *GmPRR* gene was mapped on scaffold-32, Chr04 and Chr06 each contained two *GmPRRs*, and the other nine chromosomes showed one *GmPRR*, respectively. They were renamed from *GmPRR1* to *GmPRR14* according to their chromosomal positions (Appendix A). Although *GmPRR9* and *GmPRR10* do not have a CCT domain, we retained them for further analysis, because they have been demonstrated to regulate soybean photoperiodic flowering [25,26,27]. Further synteny analysis showed fragmental duplication among *GmPRRs*, suggesting a functional similarity among *GmPRR* members (Appendix A).

Gene characteristics, including the length of the protein sequence, the protein molecular weight, isoelectric point, and the subcellular localization, were analyzed (Appendix A). The length of the 14 GmPRR proteins ranges from 558 (*GmPRR6*) to 765 (*GmPRR9*) amino acids, with molecular weights from 62.2 to 83.2 kDa. The predicted isoelectric points varied from 5.55 (*GmPRR4*) to 8.01 (*GmPRR11*). The predicted subcellular localization results showed that all of the GmPRR proteins were located in the nucleus, corresponding to Arabidopsis PRRs encoding transcription factors.

### 2.2. Evolutionary Analysis of the PRR Gene Family in Soybean

To explore the evolutionary relationships of PRR proteins, a phylogenetic tree was constructed using the whole protein sequences of *GmPRRs*, *APRRs* and *PRRs* of other crops. The results showed that *GmPRR* homologous genes were clustered into three groups, including the PRR1, PRR3/7 and PRR5/9 groups based on their similarity to the respective Arabidopsis and rice proteins (Figure 1). The PRR1 group contained four GmPRRs (GmPRR2, 4, 6 and 13), the PRR3/7 group consisted of four GmPRRs (GmPRR8, 9, 10 and 11), and the PRR5/9 group contained six GmPRRs (GmPRR1, 3, 5, 7, 12 and 14). In addition, *GmPRR2* and *GmPRR3*, and *GmPRR5* and *GmPRR6* on the same chromosome were not divided into the same group, indicating evolutionary divergence and functional diversity of *GmPRRs* on the same chromosome (Figure 1 and Appendix A). 

### 2.3. Gene Structure, Motif Composition and Promoter Characterization of Soybean PRR Gene Family

The architecture of *GmPRR* genes was examined to gain more insight into the evolution of the PRR family in soybean. The results showed that the gene length and structure were diverse among the three groups, but genes in the same group tended to share similar exon and intron structures (Figure 2). The gene length of the second group is longer than the first and third groups due to the longer intron length. Genes in Group I and Group III had a similar gene length. The 14 *GmPRR* genes possessed six to nine exons, and all of the genes in Group I contained eight exons, indicating that different *GmPRR* genes have diverged structurally during evolution. The conserved motifs of the GmPRR proteins were analyzed using Multiple Em for Motif Elicitation (MEME) (Figure 2C). Three motifs (Motif1, 3 and 4) constituted the PR domain, and were conserved in the *GmPRRs*. The Motif6 and Motif10 were the conserved Motifs of the CCT domain except for *GmPRR9* and *GmPRR10*. Among them, the first group is unique to Motif 9 and only existed in the *GmPRRs* of Group I, and Motif 8 was only shared in members of Group III. Taken together, the conserved Motifs and similar gene structures of the GmPRR members in the same group strongly support the reliability of the group classifications. These results suggest an evolutionary divergence among *GmPRR* members, and also the functional similarity of *GmPRRs* in the same group. 

Cis-acting regulatory elements (CAREs) play a vital role in the regulation of gene expression, so a cis-elements analysis was conducted using the 2 kb sequence upstream of the start codon (ATG) of the *GmPRR* genes (Appendix A). A large number of cis-elements, including light-responsive elements (I-box, Sp1, GT1-motif, 3-AF1 binding site, AE-box, chs-CMA1a, chs-CMA2a, GA-motif, Box II, LAMP-element, GATT-motif, G-box, ACE, MRE, Box 4), circadian rhythm element (circadian), defense and stress response elements (TC-rich repeats) and hormone response elements (TGA-element, TATC-box, P-box) were identified. These results suggest that *GmPRRs* may play an important role in the photoperiod response and circadian clock in soybean.

### 2.4. The Expression Pattern of GmPRR Genes

For investigating the expression patterns of *GmPRR* genes, a qRT-PCR analysis was carried out. *GmPRRs* were expressed widely in the trifoliolate leaf, unifoliolate leaf, shoot apex, stem, hypocotyl, and root of Zigongdongdou (ZGDD) and Heihe 27 (HH27) (Appendix A). *GmPRR2* and *GmPRR6* were expressed in all tissues, with higher expression in the root and lower expression in the trifoliolate leaf and unifoliolate leaf. *GmPRR4* and *GmPRR13* were more highly expressed in the trifoliolate leaf and less highly expressed in the root. The transcript levels of *GmPRR3* and *GmPRR5* were highest in the unifoliolate leaf and trifoliolate leaf, and lower in the stem and hypocotyl. *GmPRR8*, *GmPRR10* and *GmPRR11* were highly expressed in the unifoliolate leaf and the expression of *GmPRR9* was higher in unifoliolate and trifoliolate leaves. The expression of *GmPRR1*, *GmPRR7*, *GmPRR12* and *GmPRR14* was higher in the unifoliolate and trifoliolate leaves but lower in the root and shoot apex.

The leaf is the major organ for sensing the photoperiod. To evaluate the circadian clock properties of *GmPRRs*, we examined the expression of *GmPRRs* in trifoliolate leaves of ZGDD, a photoperiod-sensitive soybean variety under long-day (LD) and short-day (SD) treatments. All the *GmPRRs* showed obvious rhythmic expression except for *GmPRR12* under LD and SD conditions (Figure 3). The transcripts of *GmPRR1*, *GmPRR7*, *GmPRR9*, *GmPRR10* and *GmPRR14* peaked at 8 h after exposure to light. *GmPRR2*, *GmPRR4*, *GmPRR6*, *GmPRR8*, *GmPRR11* and *GmPRR13* peaked at 12 h after exposure to light. *GmPRR3* and *GmPRR5* transcripts peaked at 8 h under SD conditions and 12 h under LD conditions. We also found that *GmPRRs* within the same group exhibit similar rhythmic expression pattern. Except for *GmPRR3* and *GmPRR5*, *GmPRR* genes showed a higher expression level under LD conditions, suggesting that LD induced their expression in leaves. These results indicate that *GmPRRs* may regulate soybean circadian rhythm and photoperiodic flowering.

### 2.5. Haplotype Analysis of the 14 Soybean PRR Family Genes in Soybean Germplasm with Diverse Geographical Origins

In an attempt to evaluate the effect of natural variations in *GmPRR* alleles on soybean adaptation, the genotypes of *GmPRRs* were detected among 207 resequencing soybean varieties (Appendix A), which have been traditionally planted in China and the USA (Appendix A). We also investigated the flowering time across six environments (Sanya 2016, Xiangtan 2017, Xinxiang 2016, Beijing 2016, Changchun 2017 and Heihe 2017) and the geographical distribution of varieties with major haplotypes (Figure 4 and Figure 5). By comparing the flowering times of different *GmPRR* haplotypes in the six environments of China, we found that natural variations in all of the *GmPRRs* were associated with soybean flowering time and the phenotypic difference tendency becomes more pronounced with increasing latitude in China (Figure 4 and Figure 5).

*GmPRR1* sequence comparisons identified nine polymorphic loci, including seven SNPs and two indels, and defined five haplotypes (Appendix A). *GmPRR1^H3^* carried a missense mutation resulting in an amino acid substitution (Ser/Pro) outside of the PR and CCT domains (Appendix A). *GmPRR1^H3^* and *GmPRR1^H4^* were widely distributed across China and the US with similar flowering times (Figure 4A and Figure 5A). *GmPRR1^H1^* showed significantly earlier flowering (Figure 4A) and was distributed in Northeast China (NE) (Figure 5A). *GmPRR1^H5^* showed significantly later flowering (Figure 4A) and was all distributed in the US germplasm (Figure 5A). Further characterization of these two unique haplotypes (*GmPRR1^H1^* and *GmPRR1*^H5^) might facilitate soybean genetic improvement in China and the US. Seven SNPs and six haplotypes were identified for *GmPRR2* (Appendix A). SNP-Chr04:41757081 was located in the PR domain, resulting in amino acid substitution (Gln/His) (Appendix A), which may lead to late flowering for *GmPRR2^H6^* (Figure 4B). Except for Sanya2016, *GmPRR2^H1^* resulted in significantly earlier flowering, whereas *GmPRR2^H5^* and *GmPRR2^H6^* displayed later flowering (Figure 4B). *GmPRR2^H4^* was the most abundant. With the increase in latitude of China, the proportion of *GmPRR2^H1^* increased, whereas the proportion of *GmPRR2^H3^*, *GmPRR2^H4^* and *GmPRR2^H5^* decreased (Figure 5B).

For *GmPRR3*, seven SNPs and five haplotypes were defined, and SNP-Chr04:49760646, was a missense mutation (Cys/Ser) (Appendix A). *GmPRR3^H3^* and *GmPRR3^H4^* flowered significantly later than *GmPRR3^H1^*(Figure 4C). The frequency of *GmPRR3^H1^* increased with increasing latitude in China (Figure 5C). *GmPRR3^H4^* only existed in South China (SC), indicating that *GmPRR3^H4^* was strongly selected in low-latitude regions.

Eight SNPs were found, and four haplotypes were defined in *GmPRR4* and SNP-Chr05:2178871, a missense mutation resulting in amino acid substitutions (Leu/Ser) (Appendix A). Compared with *GmPRR4^H1^*, *GmPRR4^H2^* flowered significantly later (Figure 4D). *GmPRR4^H1^* was mainly distributed in NE, while *GmPRR4^H2^* was mainly distributed in Huang-Huai-Hai (HHH) and SC (Figure 5D).

Among the 16 SNPs in *GmPRR5*, two led to amino acid substitution sites, and SNP-Chr06:11184185(Glu/Asp) was in the PR domain (Appendix A). *GmPRR5^H2^* and *GmPRR5^H4^* flowered significantly later in Beijing and Changchun (Figure 4E) and were mostly found in HHH and SC (Figure 5E). *GmPRR5^H1^* flowered significantly earlier in all environments and the frequency of *GmPRR5^H1^* increased with increasing latitude in China, indicating that *GmPRR5^H1^* was strongly selected at high latitudes during soybean improvement, especially in NE (Figure 5E).

One indel and 15 SNPs were identified corresponding to five haplotypes for *GmPRR6*. SNP-Chr06:17610272 was a missense mutation (Leu/Ser) located in the PR domain (Appendix A). More than half of the varieties in the NE belong to *GmPRR6^H1^* with significantly earlier flowering (Figure 4F and Figure 5F). *GmPRR6^H2^* was mostly distributed in middle and high latitudes (Figure 5F). *GmPRR6^H3^* and *GmPRR6^H4^* were all associated with later flowering, the frequencies of which increased with decreasing latitude (Figure 4F and Figure 5F). These results revealed that diverse *GmPRR6* haplotypes with varied flowering times have adapted to target regions during soybean breeding.

For *GmPRR7*, one indel and 15 SNPs were detected, and five caused a missense mutation (Appendix A). Among the four haplotypes, *GmPRR7^H3^* was the most widely distributed across China and the US (Figure 5G), while *GmPRR7^H4^*, associated with significantly later flowering, only existed in HHH and SC (Figure 4G and Figure 5G).

Based on 17 SNPs, *GmPRR8* was divided into three haplotypes (Appendix A). The percentage of *GmPRR8^H1^* was high in NE, and *GmPRR8^H2^* and *GmPRR8^H3^* showed later flowering and were mainly found in HHH and SC (Figure 4H and Figure 5H). 

Four haplotypes were defined for *GmPRR9* according to one indel and 16 SNPs (Appendix A). *GmPRR9^H1^* carried a frameshift mutation that resulted in the loss of the CCT domain in the encoded protein (Appendix A). Compared with *GmPRR9^H2^*, *GmPRR9^H3^* and *GmPRR9^H4^*, *GmPRR9^H1^* flowered significantly earlier (Figure 4I), and was the most widely distributed across China and the US (Figure 5I). The frequency of *GmPRR9^H1^* increased with increasing latitude in China, whereas the percentages of *GmPRR9^H2^*, *GmPRR9^H3^* and *GmPRR9^H4^* decreased (Figure 5I). These results suggested that *GmPRR9* might function as a flower repressor in soybean, and the large effect mutation causing the deletion of the CCT domain, might result in early flowering.

We found one indel and five SNPs in *GmPRR10* and defined three haplotypes (Appendix A). *GmPRR10^H1^* carried a nonsense mutation that causing deletion of the CCT domain (Appendix A). *GmPRR10^H1^* was significantly associated with earlier flowering than *GmPRR10^H2^* and *GmPRR10^H3^* (Figure 4J). *GmPRR10^H1^* was the most widely distributed, and the frequency of *GmPRR10^H1^* strongly increased with increasing latitude in China, whereas the frequencies of *GmPRR10^H2^* and *GmPRR10^H3^* decreased (Figure 5J). Taken together, *GmPRR9* and *GmPRR10* both contained large effect mutations causing the loss of the CCT domain, which may disrupt their function in the regulation of flowering time in soybean.

For *GmPRR11*, 17 SNPs were found and SNP-Chr13:24842201, SNP-Chr13:24842656 and SNP-Chr13:24844364 were missense mutations resulting in amino acid substitutions (Appendix A). Two haplotypes were identified, and *GmPRR11^H1^*, which flowered significantly earlier, was widely distributed at high latitudes, while *GmPRR11^H2^* was distributed at low latitudes (Figure 4K and Figure 5K).

Based on 14 SNPs and one indel, five haplotypes were found in *GmPRR12* (Appendix A). Four SNPs were missense mutations, and SNP-Chr16:1602541 was located at the CCT domain. Compared with the other three haplotypes, *GmPRR12^H4^* was significantly later flowering and was only distributed in HHH and SC (Figure 4L and Figure 5L). These results indicate that the missense mutation in the conserved CCT domain might disrupt the function of *GmPRR12*.

Among the six SNPs, one indel was found in *GmPRR13* (Appendix A), and SNP-Chr17:8022010 was a missense mutation resulting in amino acid substitutions (Ile/Thr). Four haplotypes were identified, and *GmPRR13^H3^* was the most widely distributed. *GmPRR13^H1^* had significantly earlier flowering in Changchun and Heihe (Figure 4M) and varieties with *GmPRR13^H1^* were distributed at higher latitudes in China (Figure 5M).

Fifteen SNPs and one indel were found for *GmPRR14*, and SNP-Chr19:50366034 was a missense mutation (Phe/Ser) in the PR domain (Appendix A). Based on the allele polymorphism, a total of five haplotypes were identified, of which *GmPRR14^H2^* was the most widely distributed. *GmPRR14^H1^* flowered significantly earlier in Changchun and Heihe (Figure 4N) and appeared only in the NE varieties (Figure 5N).

### 2.6. Haplotype Combinations of GmPRRs in 207 Soybean Varieties 

To evaluate the combinatorial effects of *GmPRRs* on the local fitness of soybean, we analyzed the haplotype combinations of *GmPRRs* among the 207 soybean varieties. A total of 150 haplotype combinations were confirmed, which suggested that *GmPRR* family members were highly diversified in cultivated soybean (Appendix A). Soybean varieties harboring more haplotypes of *GmPRRs* with early flowering time tended to distribute in higher latitude regions, while varieties harboring more haplotypes of *GmPRRs* with later flowering time were mostly cultivated in the lower latitudes of China (Figure 5, Appendix A). These findings substantially coincided with haplotype analysis for individual *GmPRR* family genes (Figure 4 and Figure 5). Thus, we speculate that this diverse allelic combination of *GmPRRs* contributed to the regional adaptability of soybean. Further, we investigated haplotype combinations of *GmPRRs* in a widely-grown soybean variety of Zhonghuang 13, which ranked No.1 in planting area in the first two decades of the 21st century in China [42,43,44]. The result showed Zhonghuang 13 harbors *GmPRR1^H4^*, *GmPRR2^H4^*, *GmPRR3^H3^*, *GmPRR4^H2^*, *GmPRR5^H4^*, *GmPRR6^H4^*, *GmPRR7^H3^*, *GmPRR8^H2^*, *GmPRR9^H1^*, *GmPRR10^H1^*, *GmPRR11^H1^*, *GmPRR12^H4^*, *GmPRR13^H2^* and *GmPRR14^H2^* of *GmPRR* genes, and other varieties carrying these haplotypes, except *GmPRR5^H4^* and *GmPRR12^H4^,* are distributed in all three regions (NE, SC and HHH) in China (Figure 5, Appendix A). An association analysis revealed that *GmPRR7^H3^*, *GmPRR9^H1^*, *GmPRR10^H1^* and *GmPRR11^H1^* in Zhonghuang 13 were associated with earlier flowering; *GmPRR3^H3^*, *GmPRR4^H2^*, *GmPRR5^H4^*, *GmPRR6^H4^* and *GmPRR12^H4^* were related to later flowering; and the other haplotypes (*GmPRR1^H4^*, *GmPRR2^H4^*, *GmPRR13^H2^* and *GmPRR14^H2^*) were associated with a medium flowering time (Figure 4). These findings might partly explain the wide adaptability of Zhonghuang 13. Taken together, various combinations of mutations at *GmPRRs* provided extensive genetic plasticity that may contribute to soybean cultivated across diverse latitudes.

## 3. Discussion

*PRR* family genes play an essential role in maintaining circadian clock stability and affect plant growth and developmental processes, such as flowering time, photosynthesis response, heat shock response, oxidative stress response, stomatal conductance, mitochondrial metabolism, and cold stress [45,46,47]. In this study, a total of 14 soybean *PRR* genes were identified on 12 chromosomes (Appendix A). Phylogenetic analysis classified the 14 *GmPRR* genes into three main groups (*PRR1*/*TOC1*, *PRR3*/*7*, *PRR5*/*9*) (Figure 1), which was in accordance with PRR proteins in model plants, such as Arabidopsis and rice [31,33,48]. The number of *PRR* genes in soybean exceeds that in Arabidopsis and rice, so we speculated that soybean *PRR* genes may have a higher duplication rate or a lower gene loss rate after duplication (Appendix A) [49].

*GmPRRs* have both the N-terminal response-regulator receiver domains and the C-terminal CCT domain, except for *GmPRR9* and *GmPRR10*, whose proteins from reference genome W82 carry large effect mutations causing the loss of the CCT domain. This is consistent with previous studies [24,25,26]. qRT-PCR analysis (Figure 3) revealed that the transcripts of 13 *GmPRRs* exhibit diurnal patterns in leaves that peaked 8 h, 12 h, and 14 h after the lights were turned on, indicating that *GmPRR* expression is modulated by the circadian clock. These findings will facilitate the characterization of the circadian clock in soybean.

In Arabidopsis, the PR and CCT domain of *APRRs* confer repression activity and DNA-binding activity, respectively [50,51]. In rice, missense mutations occurring in the conserved CCT domain, are predicted to affect *OsPRR37* function in the regulation of photoperiod sensitivity, leading to an early heading date [36,38]. In this study, *GmPRR12^H4^* carried a missense mutation in the CCT domain and was significantly late flowering (Appendix A, Figure 4L). Additionally, *GmPRR3a* and *GmPRR3b*, corresponding to *GmPRR9* and *GmPRR10* in this study [25] (Appendix A, Appendix A), respectively, carry mutations that delete the CCT domain in the encoded proteins leading to early flowering in soybean [24,25,26]. Thus, it is worth clarifying the function of *GmPRR12* in the photoperiodic flowering pathway. However, non-synonymous variants in the PR domain appeared in *GmPRR2*, *GmPRR5*, *GmPRR6*, *GmPRR7*, and *GmPRR14*, and only the variation in *GmPRR2* was clearly associated with flowering time (Figure 4). Further studies could verify the effect of the PR domain on the regulation of soybean photoperiod flowering. For *GmPRR1*, *GmPRR4*, *GmPRR8*, *GmPRR11* and *GmPRR13*, there were some missense variants occurring in the coding regions outside the CCT and PR domains, which were associated with flowering time. These variations might weakly affect gene function and finely tune the developmental rate. Developing Kompetitive Allele Specific PCR (KASP) markers for these variations could facilitate soybean improvement with the desired flowering time for planting in target areas.

The circadian clock coordinates the internal biological processes with the external environmental factors, and thus provides an adaptive advantage. The occurrence of circadian timekeeping is of great importance to living beings. Studies have shown that naturally occurring variations in clock parameters (period, phase, and amplitude) are necessary for the circadian clock to contribute to the fitness of organisms over a wide range of latitudes [52,53,54,55,56,57]. In Arabidopsis thaliana, latitude-specific selection effects have been found on circadian properties by analyzing natural variation in the period, phase, and amplitude of 150 accessions, and *Pseudo-response regulator* (*PRR*) family members are key candidates for clock quantitative trait loci [52]. Crops are commonly cultivated over a broader geographical range compared with their ancestors and increasing evidence has shown that breeders have been indirectly selecting for circadian parameters, which might contribute to improved performance in distinct latitudes. The allelic variation of *EID1* responsible for the phase delay was selected in cultivated tomato during domestication due to the enhanced performance under long-day photoperiods [53]. Similarly, latitudinal clines in circadian period were found in elite soybean cultivars from six maturity groups [54]. Thus, further analysis of the association between the genetic diversity and circadian period of *GmPRRs* will facilitate elucidating the adaptation of soybean to different cultivated regions.

Unlike other traits that generally experience breeding selection in one direction, soybean flowering time becomes more diverse during breeding for different environments with variable day lengths. It has been a major goal for breeders to decipher the genetic mechanisms of soybean flowering time and regional adaptability. In this study, we analyzed the distribution of major *GmPRR* haplotypes among 180 varieties traditionally cultivated in China. Soybean varieties with *GmPRR1^H1^*, *GmPRR2^H1^*, *GmPRR3^H1^*, *GmPRR4^H1^*, *GmPRR5^H1^*, *GmPRR6^H1^*, *GmPRR7^H2^*, *GmPRR7^H3^*, *GmPRR8^H1^*, *GmPRR9^H1^*, *GmPRR10^H1^*, *GmPRR11^H1^*, *GmPRR12^H1^*, *GmPRR12^H2^*, *GmPRR12^H3^*, *GmPRR13^H1^*, and *GmPRR14^H1^* were associated with earlier flowering and tend to be distributed in higher latitudes in China (Figure 4 and Figure 5). However, varieties of *GmPRR1^H5^*, *GmPRR2^H5^*, *GmPRR2^H6^*, *GmPRR3^H3^*, *GmPRR3^H4^*, *GmPRR4^H2^*, *GmPRR5^H2^*, *GmPRR5^H4^*, *GmPRR6^H3^*, *GmPRR6^H4^*, *GmPRR7^H4^*, *GmPRR8^H2^*, *GmPRR8^H3^*, *GmPRR9^H2^*, *GmPRR9^H3^*, *GmPRR9^H4^*, *GmPRR10^H2^*, *GmPRR10^H3^*, *GmPRR11^H2^*, *GmPRR12^H4^*, *GmPRR13^H3^* and *GmPRR14^H4^*, which were related to later flowering, are mainly found in Huang-Huai-Hai (HHH) and South China (SC) (Figure 4 and Figure 5). We found high diversity of haplotype combinations of *GmPRRs* in the 207 accessions, and the combinational effects of *GmPRRs* may contribute to the wide adaptation of Zhonghuang 13 (Figure 5, Appendix A). Furthermore, these rich natural variations and combinations of *GmPRRs* may be useful for precise prediction of flowering time.

Soybean flowering time is a quantitative feature regulated by various genes [8,9,10,11,12,13,14,15,16,17,18,19,20,21,22,23,24,25,26,27,28,29]. Recent studies have shown that clock genes play important roles in soybean flowering and ecological adaptation. Natural mutants of *GmELF3*/*J* [20,21,22] and *LHY1a* [27] improve soybean adaptation to the tropics, and natural variations in *GmPRR37*/*GmPRR3b* and *GmPRR3a* are associated with soybean adaptation to high latitudes [24,25,26]. GmLUX1 and GmLUX2 both interact with GmELF3/J to form an evening complex and play essential roles in soybean flowering and adaptation [22]. Thus, we also investigated the number of homologs for clock genes including *CCA1*/*LHY*, *LNK* family, *ELF3*, *ELF4*, and *LUX*, and found that the number of these genes (CCA1/LHY-4, LNK-4, ELF3-5, ELF4-2, LUX-2 homologs) are much fewer than *PRR* genes in soybean. Further analysis of the allelic combinations of *GmPRRs* with other genes controlling flowering would allow us to precisely manipulate flowering time, and to breed soybean varieties best adapted to diverse environments.

## 4. Materials and Methods

### 4.1. Plant Materials, Photoperiod Treatments and Multiple-Site Experiments

For the analysis of the expression patterns of *GmPRRs*, the photoperiod sensitive soybean variety Zigongdongdou (ZGDD) and the photoperiod insensitive soybean variety Heihe27 (HH27) were grown in a controlled culture room at 26 °C under short-day (SD) (12 h: 12 h, light: dark) and long-day (LD) (16 h: 8 h, light: dark) conditions. Different organs of the plant, including the trifoliolate leaf, unifoliolate leaf, shoot apex, stem, hypocotyl, and root were sampled after 4 h exposure to light at 10 days after emergence (DAE). The trifoliolate leaves of soybean variety ZGDD were sampled at 4 h intervals throughout a 48 h period on days 10 and 11 of LD or SD treatment. Each sample consisted of material collected from three plants.

A 207-accession panel including 180 soybean varieties from China (97 from Northeast China (NE), 46 from Huang-Huai-Hai (HHH), 37 from South China (SC)) and 27 from the US were used for haplotype analysis (Appendix A) [58]. The panel was planted in Sanya (18°18′ N, 112°39′ E), Xinxiang (35°08′ N, 113°45′ E)) and Beijing (40°13′ N, 116°33′ E) in 2016, and in Xiangtan (27°40′ N, 112°39′ E), Changchun (43°50′ N, 124°82′ E) and Heihe (50°24′ N, 127°49′ E) in 2017 [59]. These six environments were named SY2016, BJ2016, XX2016, XT2017, CC2017 and HH2017, respectively. All of the 207 soybean varieties were grown in rows 1.5 m long by 0.5 m row space with 0.1 m between the plants. All materials were arranged in randomized complete blocks with two replications. 

### 4.2. Identification, Phylogenetic and Bioinformatic Analysis of GmPRRs

The potential PRR family members in soybean were identified based on Arabidopsis PRR sequences using both BLASTp and Hidden Markov Model (HMM). The protein sequences of Arabidopsis PRRs were downloaded from the TAIR database (https://www.arabidopsis.org/) (accessed on 8 February 2020). Soybean genome annotation was downloaded from the Phytozome 13.0 database (https://phytozome-next.jgi.doe.gov/) (accessed on 8 February 2020). Soybean PRR members resulting from both searches with an E-value threshold of <e^−10^ were pooled, and all redundant putative PRR sequences were removed. The remaining PRR sequences were further confirmed by the Pfam (http://pfam.xfam.org/) (accessed on 12 February 2020) and NCBI Conserved Domains Database (https://www.ncbi.nlm.nih.gov/cdd/) (accessed on 12 February 2020) for the existence of a PR domain and CCT domain.

The Multiple Collinearity Scan toolkit (MCScanX) was used to analyze the *GmPRRs* duplications with the default parameters (MCScanX: a toolkit for detection and evolutionary analysis of gene synteny and collinearity). The syntenic analysis map was constructed using TBtools [60]. Physiochemical parameters of soybean PRR proteins, such as polypeptide length, the protein molecular weight, isoelectric point was investigated using ExPASy (http://web.expasy.org/protparam/) (accessed on 15 September 2021). The subcellular localization of GmPRR proteins was predicted by PSORT II (https://psort.hgc.jp/form2.html) (accessed on 15 September 2021).

To reveal the evolutionary relationships among PRR genes in different plant species, potential PRR genes from Arabidopsis, rice and other crops were selected for phylogenetic analysis. The phylogenetic tree was constructed using the Neighbor-Joining (NJ) method of MEGA 7.0 software (https://www.megasoftware.net/home) (accessed on 8 February 2020) with a bootstrap test of 1000 replicates [61].

### 4.3. Gene Structure, Conserved Motif and Promoter Sequence Analyses 

For the analysis of the exon–intron patterns, the genomic DNA sequence and CDS of each *GmPRR* gene were retrieved from the soybean genome database. Conserved motifs of *GmPRRs* were analyzed by the Multiple Em for Motif Elicitation (MEME, https://meme-suite.org/meme/tools/meme) (accessed on 26 May 2021), and the maximum number of motifs was set to 10. 

To identify the potential photoperiod and circadian related cis-elements, the 2 kb genomic sequences upstream of the start codon (ATG) of *GmPRR* genes were extracted from the soybean genome database. The cis-acting elements were confirmed using Plant Cis-Acting Regulatory Element (PlantCARE, http://bioinformatics.psb.ugent.be/webtools/plantcare/html/) (accessed on 26 May 2021). TBtools software (https://github.com/CJ-Chen/TBtools) (accessed on 30 May 2021) was used for visualization of these analysis results [60].

### 4.4. RNA Extraction and Quantitative Real-Time PCR

Total RNA was extracted using the Easy Fast Plant Tissue Kit (TianGen, Beijing, China), and cDNA was synthesized with the FastKing RT Kit (With gDNase) (TianGen, Beijing, China). Primers were designed using the NCBI (https://www.ncbi.nlm.nih.gov/tools/primer-blast/) (accessed on 13 May 2021) online program, and the primers (Appendix A) were synthesized by Tsingke Biotechnology Co., Ltd. (Beijing, China). Quantitative RT-PCR (qRT-PCR) was performed using an ABI QuantStudio™ 7 flex (Applied Biosystems, San Francisco, CA, USA) with Taq Pro Universal SYBR qPCR Master Mix (Vazyme, Beijing, China). Three biological replicates of every sample were measured. The reaction procedures were as follows: pre-denaturation at 95 °C for 30 s; denaturation at 95 °C for 5 s, annealing at 60 °C for 30 s, 40 cycles. Each reaction was performed in biological triplicates and the relative gene expression levels were analyzed using 2^−^^ΔΔCT^ method with the *GmActin* (*Glyma18g52780*) gene as an internal control.

### 4.5. Haplotype Analysis of Soybean PRR Genes

The 207 whole-genome resequencing soybean varieties were obtained from our previous study [58] (Appendix A), and the sequence data have been deposited in the NCBI database under Short Read Archive (SRA) Accession Number SRP062560 and PRJNA589345. The SNPs and insertion and deletion polymorphism were extracted for further haplotype analysis. The flowering time across six environments of 207 soybean varieties was recorded as days from emergence to beginning to bloom [62]. The flowering time phenotype of each variety in a single environment was determined by taking the average from two replications. The association analysis of *GmPRR* haplotypes with flowering time was determined by Duncan’s multiple range test using GraphPad Prism 8 (*p* < 0.05). 

## Figures and Tables

**Figure 1 ijms-23-09970-f001:**
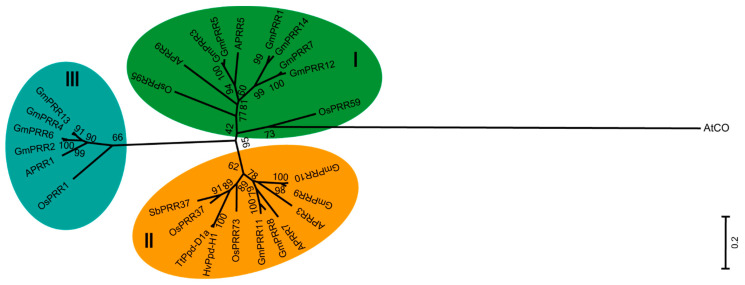
Phylogenetic tree of PRRs in higher plants. Different groups are represented by different colors. I, the PRR5/9 group; II, The PRR3/7 group; III, The PRR1 group. The phylogenetic tree was constructed by MEGA 7.0 using the Neighbor-Joining (NJ) method with 1000 bootstrap repeats. The numbers on the branches indicate the bootstrap values. The Arabidopsis *CONSTANS* (*AtCO*) gene containing a CCT (*CONSTANS*, *CO-like*, and *TOC1*) domain was considered as an outgroup. Different subfamilies are represented by different colors.

**Figure 2 ijms-23-09970-f002:**
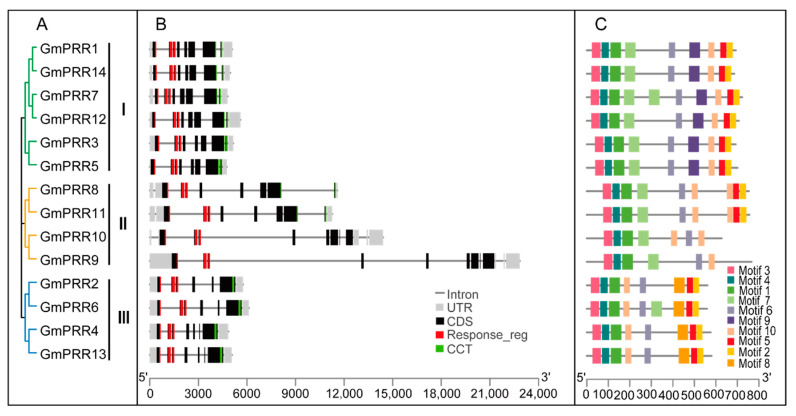
The phylogenetic relationships, gene structure, and conserved motifs of the PRR gene family in soybean. (**A**) Phylogenetic tree of soybean PRR proteins. Different subfamilies are represented by different colors. (**B**) Genetic structure of *GmPRR* genes, including introns, UTRs, CDSs and domains specific to the PRR family. (**C**) The conserved motif of GmPRR proteins and the length of each motif is shown proportionally.

**Figure 3 ijms-23-09970-f003:**
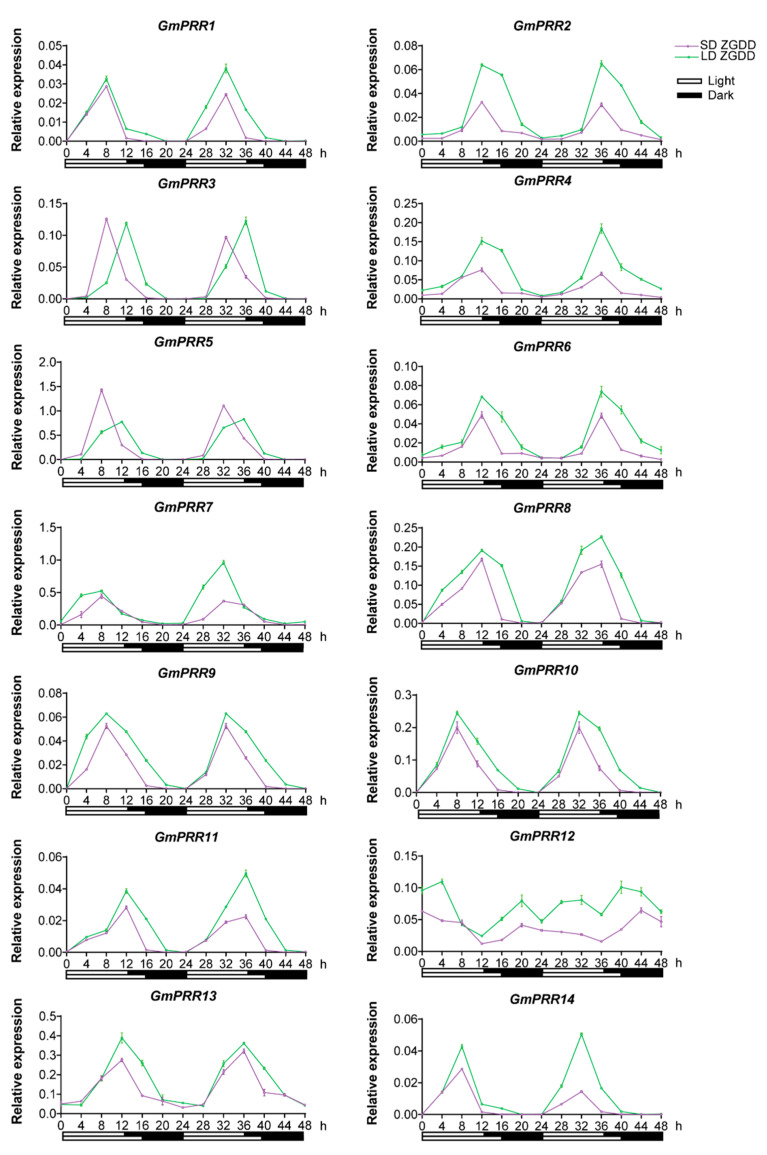
Expression levels of *GmPRRs* throughout a 48 h period in the unifoliolate leaves of soybean variety ZGDD (Zigongdongdou) on days 10 and 11 of long-day (LD, 16:8, light: dark) or short-day (SD, 12:12, light: dark) treatment. Relative transcript levels of GmPRRs were normalized to GmActin. The data are given as the means ± SE of three biological replicates.

**Figure 4 ijms-23-09970-f004:**
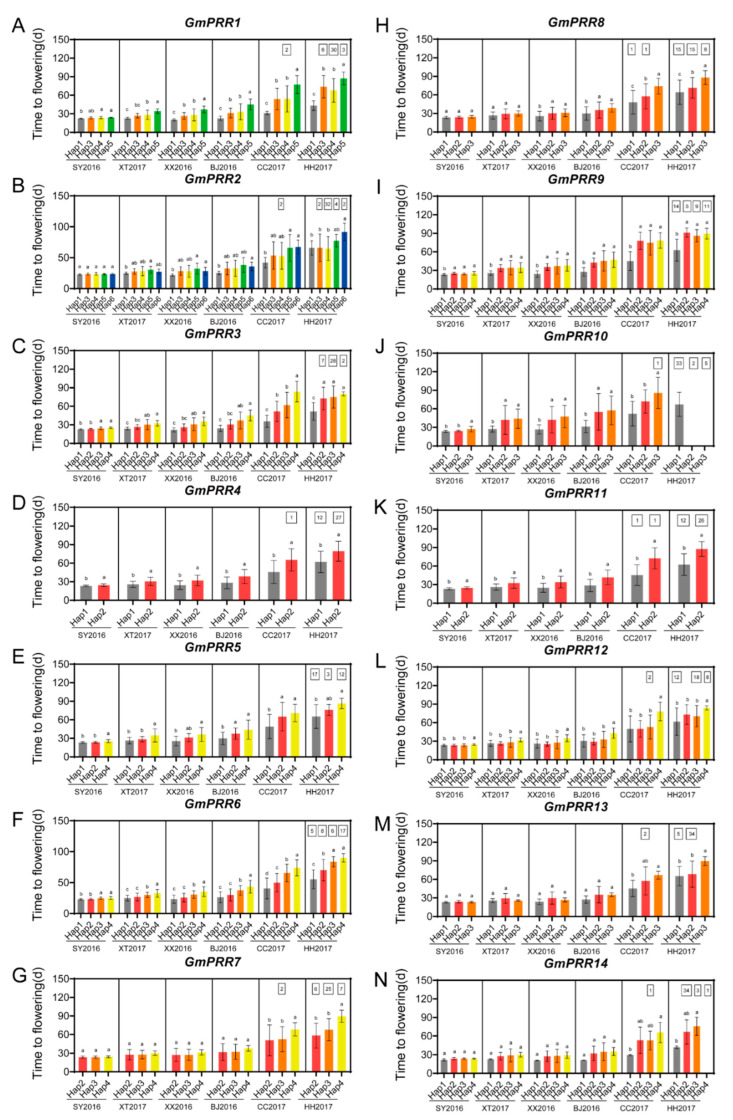
Association analysis of *GmPRR* haplotypes with flowering time in soybean germplasm. (**A**–**N**) Flowering time of the soybean varieties with major haplotypes of *GmPRR1*-*GmPRR14*. The number within each box indicates the number of varieties that did not flower. The data are means ± standard deviations, and a, b and c indicate ranking by Duncan’s test at *p* < 0.05. SY2016: Sanya2016; XT2017: Xiangtan 2017; XX2016: Xinxiang2016; BJ2016: Beijing2016; CC2017: Changchun2017; and HH2017: Heihe2017.

**Figure 5 ijms-23-09970-f005:**
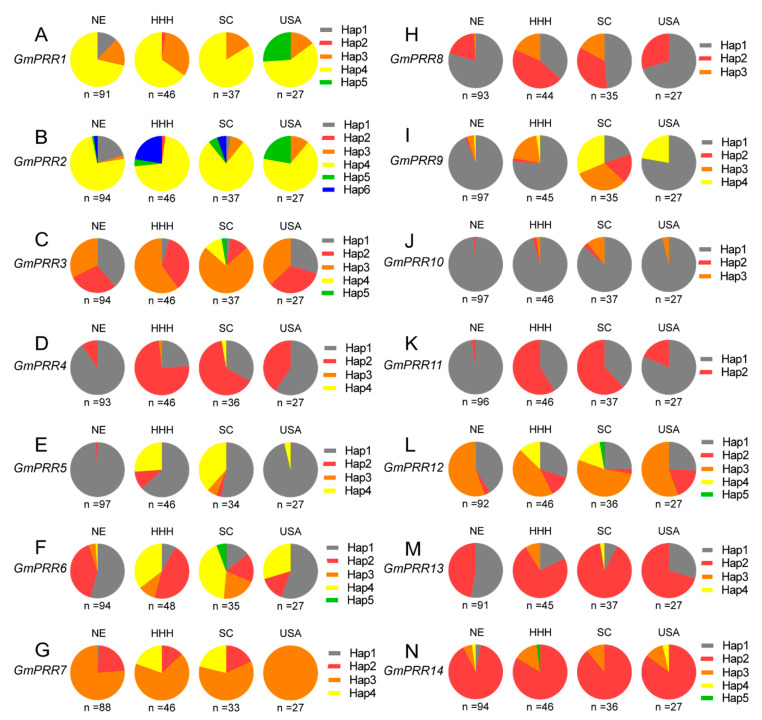
Geographical distribution of soybean varieties harboring different alleles of *GmPRR* genes. (**A**–**N**) The geographic distribution of *GmPRR1*-*GmPRR14* haplotypes. NE: Northeast China; HHH: Huang-Huai-Hai; SC: South China.

## Data Availability

The sequence data used in this study are available in the NCBI database under Short Read Archive (SRA) Accession Number SRP062560 and PRJNA589345.

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
