# Peer review of "Genomic Dissection and Diurnal Expression Analysis Reveal the Essential Roles of the PRR Gene Family in Geographical Adaptation of Soybean"

_ijms, 2022, doi:10.3390/ijms23179970_

Round 1

Reviewer 1 Report

This paper focused on the natural variation of PRR family genes, typical circadian clock genes, in soybean. The authors detected 14 PRR genes by appropriate methods and measured their expression patterns. Haplotype analysis using 207 varieties showed that haplotypes were associated with flowering time under natural conditions and the geographical distribution of cultivars. These results will be an important basis for the study of the natural variation of the soybean circadian system which is a potential target for breeding.

Major comments

#1 In some PRR genes, haplotypes showed an association between flowering time, suggesting the effect of their mutations on flowering time. However, the authors do not show direct evidence of the effect. For example, the PRR haplotypes carried by one variety, which was often used for breeding in the northern area due to its early flowering phenotype caused by the non-PRR region, will be enriched in the northern cultivars and showed association with early flowering phenotypes. I think such a case may be rare, but authors should be careful to use “caused” to describe the association. In the abstract, i.e., “especially the nonsense mutations resulting in deletion of the CCT domain, which caused early flowering.”

#2 In this paper, the mechanisms of how PRR haplotypes affect the flowering time and are geographically selected are not discussed. In Arabidopsis, the mutations in PRR genes are a potential driver for the natural variation of the circadian period (Doi: 10.1126/science.1082971). In the American soybean, circadian periods showed a latitudinal cline (DOI: 10.1177/0748730416679307). The longer circadian period was selected during the domestication of tomatoes due to the higher crop performance under long-day conditions in the northern area (DOI: 10.1038/ng.3447). Recently, in a short-day duckweed species, the correlation between the circadian period and the critical day-length on photoperiodism was reported (DOI: 10.1016/j.isci.2022.104634). Discussing the relation between PRR diversity and the natural variation of the circadian period in the introduction or discussion section would provide the idea for the selection pressure on PRRs.

#3 The evolutionary divergence of PRR genes in soybean is quite interesting. I am interested in, then, whether this divergence is unique to the PRR genes. How about other clock genes such as CCA1/LHY, LNK family, ELF3, ELF4, and LUX? The number of these genes would be important to reveal the role of PRRs in the adaptive evolution of the soybean circadian clock.

# Minor points

Abstract: “the expression of 12 GmPRRs was induced by LD in leaves.” > “the expression of 12 GmPRRs was higher in LD in leaves.”

Supplementary Figure S3: Provide the sampling time in the legend.

Reviewer 2 Report

Dear Authors,

Your publication is part of the constant search for possibilities in the field of crop plant modification. The use of the latest research tools allows you to reach your destination faster. The work is written very well, although the abundance of abbreviations used in the text makes it difficult to read. In my opinion, it is suitable for publication in its current form.

Reviewer 3 Report

In this manuscript, the authors focus on the PRR gene family soybean and try to figure out the reason for geographical adaptation. The aims of this study are clear and the design of the project is acceptable. I don’t have a negative or major comment for this manuscript. Some minor suggestions are below:

  1. Figure 1, can separate into three groups. I can find the information of three groups from LINE 6-10 of PAGE 3. Can the authors label three groups in figure 1 like figure 2? It will be easy to follow.
  2. In figure 3, how many replicate tests are in each gene?
